# Systemic impacts of disruptions at maritime chokepoints

Jasper Verschuur [1,2,3] ✉, Johannes Lumma[1] & Jim W. Hall [1]

Global trade relies on a small number of strategic passageways, so-called maritime chokepoints, which are vulnerable to disruptions. Yet, the exposure of countries to these disruptions has not been comprehensively assessed, inhibiting adequate preparedness. Here, we quantify the systemic impacts of maritime chokepoint disruptions subject to a variety of hazards, both natural and human-induced. The expected value of trade disrupted at chokepoints is estimated to be USD192 billion annually, mainly attributed to geopolitical risk at the Taiwan Strait and Suez Canal, and a combination of hazards affecting the Bab el-Mandeb Strait. We estimate the economic losses of chokepoint disruptions, due to delays, rerouting, insurance premiums and trade disruptions, to be USD10.7 billion per year, and an additional USD3.4 billion per year due to increased freight costs. In both cases, risks to the Suez Canal and Bab el-Mandeb Strait drive these losses. Countries most affected are in the vicinity of these two chokepoints, but also further away, including countries in Western Africa and Central Asia. At a time of heightened geopolitical tensions and climate change, our results help to quantify the implications of these risks and can present a useful starting point to identify resilience interventions to mitigate these threats.

Maritime transport is essential for international trade, facilitating 80% of trade volume and 50% of trade value[1,2], and key for economic development[3,4]. A large share of maritime trade crosses one or more 'maritime chokepoints', strategic places where shipping routes concentrate in often narrow passageways[5,6]. These maritime chokepoints are considered vulnerable points in the global maritime transport system given the high concentration of maritime trade value and limited route alternatives[7]. For instance, around 21% of global petroleum consumption is shipped through the Strait of Hormuz (connecting the Persian Gulf with the Gulf of Oman)[8]. In addition, a number of maritime chokepoints facilitate over 10% of global grain exports, including the Bosporus Strait, Strait of Gibraltar, Panama Canal and the Malacca Strait[6].

Maritime chokepoints are prone to multiple hazards that can disrupt shipping, including piracy, geopolitical conflict, climate extremes, terrorist attacks, and shipping accidents, among others[6].

Disruptions of these chokepoints could result in detours, rising insurance and shipping rates, delays or even firms halting operations. For instance, in 2021, the Ever Given ship ran aground in the Suez Canal, blocking it for five days[9], creating vessel queues on both sides of the canal. In 2023–2024, severe drought conditions led canal operators to implement draft and traffic restrictions at the Panama Canal, causing major vessel backlogs and detours[10]. More recently, in 2023-2025, Houthi rebels attacked ships in the Red Sea, causing a large share of vessels to take a detour around the Cape of Good Hope, resulting in production halts at factories and a spike in shipping prices[10].

These events have highlighted the importance of understanding the variety of hazards that chokepoints face, how likely they are to occur, how much trade is exposed and what the impacts may be in case of disruptions, considering possible mitigation responses, e.g. rerouting. While chokepoint disruptions materialise locally, the impacts of them are systemic, requiring analysis of the entire network

[1]Oxford Programme for Sustainable Infrastructure Systems (OPSIS), Environmental Change Institute, University of Oxford, Oxford, UK. [2]Department of Engineering Systems and Services, Faculty of Technology, Policy and Management, Delft University of Technology, Delft, The Netherlands. [3]Climate Safety & Security Centre, TU Delft Campus the Hague, Delft University of Technology, The Hague, The Netherlands. ✉e-mail: jasper.verschuur@eci.ox.ac.uk

of shipping trade links and the economies they connect. Insights into these systemic risks can help anticipate these impacts, facilitate adequate responses, and make the case for joint efforts to improve the security of trade flows through chokepoints globally.

Concerns about the security implications of maritime chokepoints have existed for a long time, especially regarding energy and food security. In response, there has been extensive analysis of the security implications of particular chokepoints or for a selected set of critical commodities (e.g., oil, grains)[6,11,12]. While insightful for these specific chokepoints and sectors, these previous analyses do not provide a general view of country trade dependencies on multiple chokepoints and across all commodities. Others have used detailed vessel movement data to quantify dependencies between incoming and outgoing vessel traffic at country ports and the maritime chokepoints utilised for this traffic[13], or used network-based approaches to identify network vulnerabilities between the Panama and Suez Canal and ports globally[7]. While these approaches provide insights into the shipping dependencies on maritime chokepoints, they cannot be used to translate this information into economic impacts for specific countries as they do not capture the volume or value of trade flows.

Only recently, an analysis by Pratson (2023)[5] attempted to bridge this gap by constructing a geospatial workflow to predict which bilateral trade flows cross one or more of the eleven maritime chokepoints analysed. As the study focuses on commercial exposure and delays associated with disruptions at key points, further investigation is required into the estimation of the probability of such disruptions and their economic consequences. Gaining insights into the probability and consequences, i.e. risk, associated with certain maritime chokepoints could help policymakers and other actors to better prepare for such disruptions and design strategies to cope with them.

In this research, we aim to quantify the systemic risks of maritime chokepoint disruptions, taking into consideration the country trade dependencies on maritime chokepoints, the likelihood of various risks materialising at individual chokepoints, and the economic consequences of these disruptions. Systemic risks in this context thus refers to the economic losses that countries may experience because of the adverse impacts associated with maritime chokepoint disruptions. The economic loss is derived by first estimating the expected value of trade disrupted (EVTD) due to chokepoint disruptions, which is a simple metric of countries' relative potential for trade disruptions.

We do this by combining the output from a global maritime transport model, the Oxford Maritime Transport Model[1], with hazard event data for 24 maritime chokepoints globally (Supplementary Fig. 1). The hazards covered in this analysis include three natural hazards (cyclones, droughts and earthquakes) and five human-induced hazards (piracy, blockages, armed conflict, terrorism and interstate conflict), all with specific data sources used (Methods).

These hazards affect chokepoint differently. Some occur frequently but with short-lived disruptions (e.g., cyclones), while others have a very low probability of occurrence but can have devastating impacts (e.g., an earthquake damaging the Panama Canal). By expressing each hazard in terms of risk, covering the likelihood of disruptions (in terms of probability), the disruption duration (e.g., days to years) and severity (e.g., full or partial trade flow disruption), we can compare the impacts across hazards and chokepoints. The associated economic losses we cover include the rerouting costs, cost of delays, reduction in toll revenues, rising insurance rates (e.g., war premiums) and cost of production interruptions. This allows us to identify the countries with the highest risk in terms of their systemic risk profile. Moreover, we estimate the global economic loss associated with container price spikes, which occur because vessels must reroute and, hence, removing capacity from the global container fleet.

## Results

### Global trade through chokepoints

The magnitude of trade flows passing the various chokepoints is driven by the bilateral trade patterns, the configuration of the maritime transport network (which determines which routes are taken to connect country pairs) and the size of this trade flow. In value terms, around 20% of global maritime trade is shipped through the Taiwan Strait and Malacca Strait, making them the most important chokepoints globally (Fig. 1a). This is driven by the large trade flows between East Asia and Europe, South Asia and the Middle East, respectively, which pass these two chokepoints. Around 15% of maritime trade value transits the Suez Canal, Dover Strait, Strait of Gibraltar and Bab el-Mandeb Strait, mainly connecting Asia with Europe. The remaining chokepoints all handle less than 10% of maritime trade value. A similar pattern is observed if looking at trade flows in volume terms (Supplementary Fig. 2), although the importance of the Taiwan Strait, given large bulk trade from Australia to China, as well as the Strait of Hormuz ( ~ 13% of global maritime volume), given oil trade out of the Middle East, increases. Trade flows often cross multiple chokepoints, resulting in the sum of the share of trade across chokepoints being larger than 100%. In fact, we estimate that each USD of maritime trade results in USD1.8 in trade through these 24 chokepoints, and each tonne of maritime trade results in 2.0 tonne of trade through maritime chokepoints.

### Country trade dependencies on chokepoints

The trade dependency of countries on various chokepoints depends critically on the trade network of the country, the share of this trade being maritime (versus air or land-based transport), and the shipping routes used to facilitate this trade. Some countries almost exclusively ship goods through a certain chokepoint, such as the Baltic countries through the Oresund Strait, Eastern European and Central Asian countries through the Bosporus Strait, and Middle Eastern countries through the Strait of Hormuz. For other countries, maritime trade flows are more dispersed and only a fraction of trade passes a maritime chokepoint (see Supplementary Fig. 3).

Figure 1b shows the dominant maritime chokepoint for each country. Here, one can identify clusters of countries that depend on a particular chokepoint. For instance, North America and the west coast of Latin America depend primarily on the Panama Canal, while the east coast of South America depends critically on the Cape of Good Hope (given large exports to Asia). In Europe, a clear split is visible, with Scandinavia and Baltic States relying on the Oresund Strait, Western and Southern Europe trading most through the Strait of Gibraltar, and Eastern European countries shipping goods through the Bosporus Strait. East Asia depends primarily on the Taiwan Strait, while for South Asian countries, the Malacca Strait is the dominant chokepoint, in both cases given large interregional trade within Asia. The dominant chokepoint for African countries is more mixed and is driven by the top maritime trade flows that go through a certain chokepoint. For instance, West African countries export mining products to Asia via the Strait of Gibraltar.

Some countries rely on one main chokepoint only, while other rely on multiple chokepoints. In Supplementary Fig. 4 we quantify the number of chokepoints through which over 5%, 10% and 25% of a country's maritime trade value transits, which we designate as major trade dependencies. China and Brazil have the largest dependencies on multiple chokepoints for the 5% threshold, while Uzbekistan is the country with the largest dependency for the 10% threshold (given the chain of chokepoints it depends on – Bosporus, Suez, Gibraltar, Bab el-Mandeb). Russia, Kazakhstan and Tajikistan depend on six chokepoints that each facilitate over 25% of their maritime trade value. On the other hand, North and South American countries, China, and countries in Western Europe have fewer than two chokepoints that facilitate over 25% of maritime trade value. These countries depend

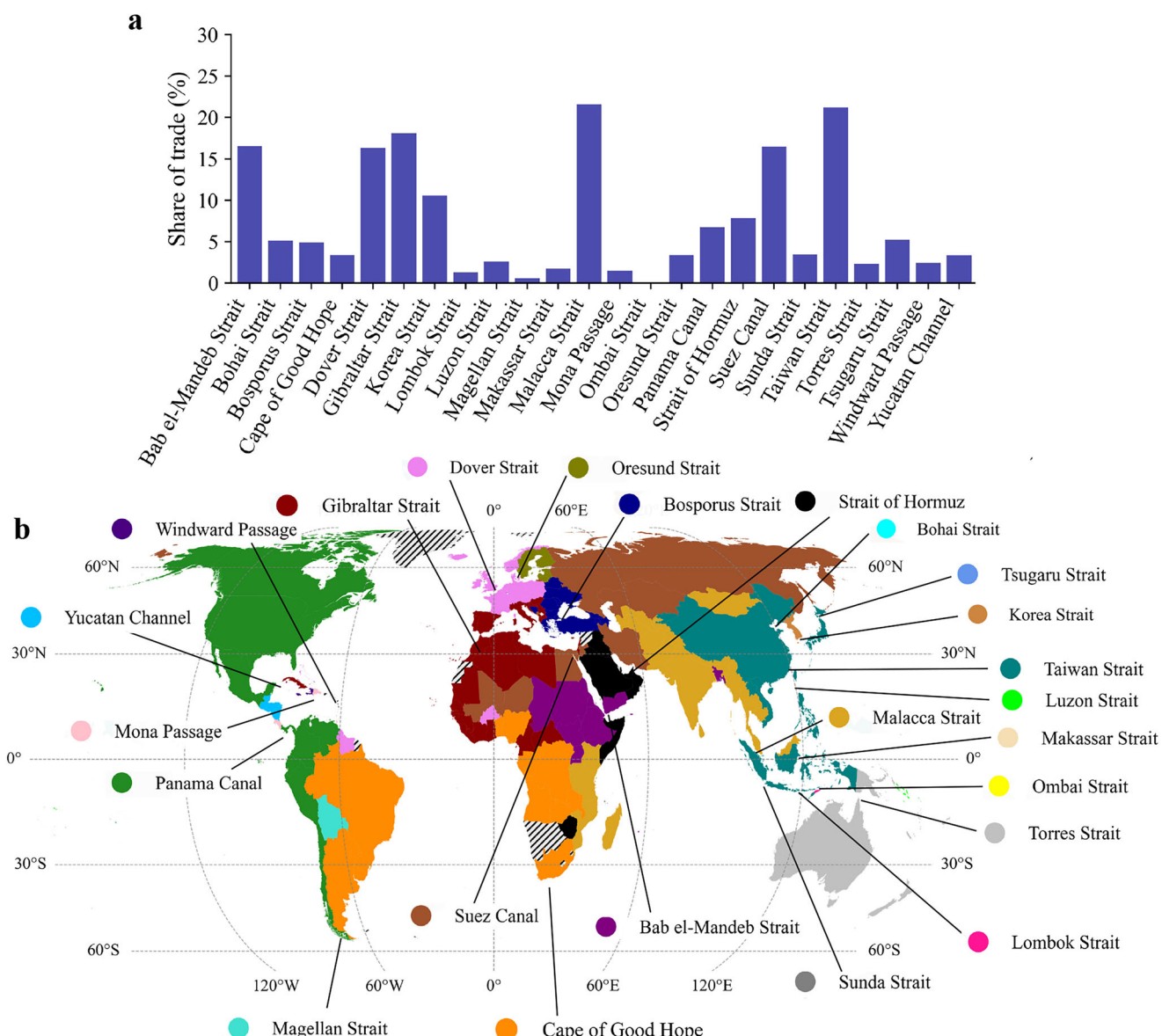

**Fig. 1 | Global and country dependencies on maritime chokepoints.** (a) The share of global maritime trade value passing the maritime chokepoints. (b) The dominant chokepoint per country in terms of trade dependency (in value terms). The hatched pattern indicates countries not included in the analysis. The exact location of the maritime chokepoints is shown in Supplementary Fig. 1. The base-map is from GADM.

primarily on chokepoint-free routes, such as from and to Western Europe and the Americas, and between China and the United States.

### Expected value of trade disrupted

The maritime chokepoints considered face potential disruptions due to a variety of hazards. We consider the following hazards at each chokepoint: cyclones, drought, earthquakes, blockages (due to accidents), terrorist attacks, armed conflict, interstate conflict and piracy. We have used a combination of empirical and model-based analysis to estimate the probability of chokepoint disruptions. This information is then combined with the trade value flowing through the chokepoints, the duration of the disruption, and the severity of the disruption (fraction of trade being affected). Taken together, we express this as the expected value of trade disrupted (EVTD), in USD per year (see Methods).

Our EVTD definition allows us to compare hazards on different ends of the risk spectrum, i.e., low probability but high impact events, as well as high probability but low impact events. Cyclones can cause

frequent disruptions to certain chokepoints (Taiwan, Korea and Tsugaru Strait), but the duration of disruption may only be a few days. Similarly, piracy risk has a relatively high frequency of occurrence at the Bab el-Mandeb Strait and the Malacca Strait but may only disrupt a fraction of trade, as only a small share of vessels may decide to avoid this chokepoint[14]. On the contrary, while interstate conflict has a low probability of occurrence, it may disrupt a certain chokepoint for a long period of time. To begin with, we treat the probabilities of each hazard as being mutually exclusive from each other at each chokepoint and assume that hazards never co-occur at different chokepoints, which generates an upper bound on the systemic risk. We further investigate this assumption later (see 'Joint chokepoint disruptions').

Globally, the EVTD is estimated to be USD191.5 billion per year, equivalent to around 0.77% of the value of global trade (USD24.9 trillion in 2022). We can disentangle this number per chokepoint and hazard combination (Fig. 2). We split the chokepoints in three categories, depending on possibility to reroute goods in case of disruptions: no rerouting, long rerouting (>5000 km) and short rerouting.

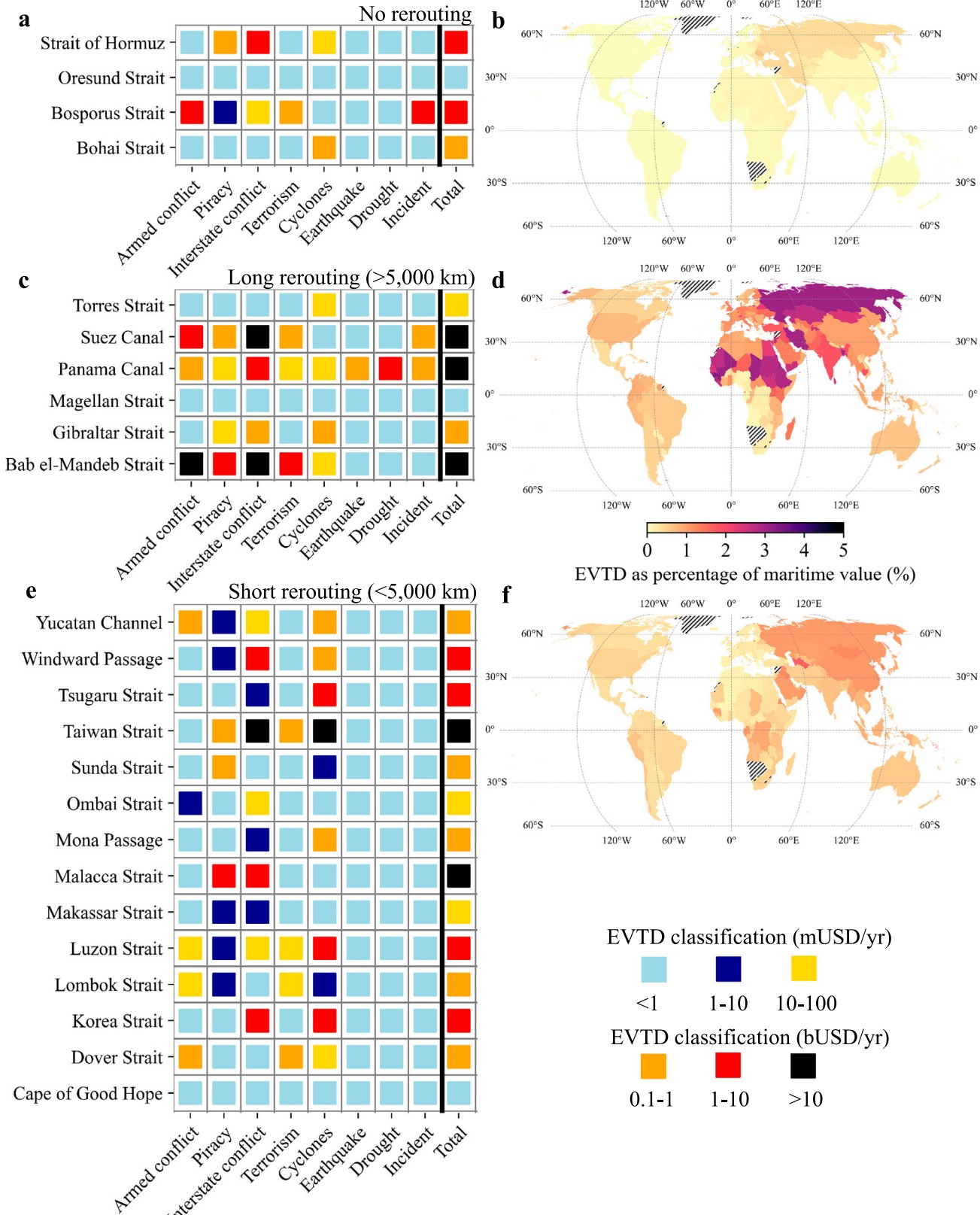

**Fig. 2 | Expected value of trade disrupted (EVTD) due to maritime chokepoint disruptions. a** The EVTD per maritime chokepoint broken down by the different types of hazards considered for chokepoints without any rerouting options. **b** The trade risk as a percentage of the total trade value of a country, highlighting countries with the largest relative trade risk. The hatched pattern indicates countries not included in the analysis. **c**, **d** same as (**a**) and (**b**) but for chokepoints that have long rerouting distances (>5000 km). **e**, **f** same as (**a**) and (**b**) but for chokepoints that have short rerouting distances (<5000 km). The basemap is from GADM.

This has implications for the economic loss that dependent countries face because of the estimated disruptions (see section 'Chokepoint economic risk').

Four chokepoints fall in the first category (Strait of Hormuz, Oresund Strait, Bosporus Strait and Bohai Strait), given that disruptions to those chokepoints cut-off countries directly from international maritime transport routes. The Bosporus Strait has the highest EVTD (USD4.1 billion per year, 0.02% of global trade), driven by the risk of blockages, domestic conflict and terrorist attacks, followed by the Strait of Hormuz (USD1.9 billion per year, 0.01% of global trade), primarily driven by the risk of geopolitical conflict (Fig. 2a). The relative EVTD as percentage of maritime trade value (Fig. 2b) is less than 1% across countries, with the highest risk countries being Romania, Azerbaijan and Ukraine, given their dependency on the Bosporus Strait. Still, the economic loss may be high, given that disruptions may translate into costly delays or firms stopping production.

Six chokepoints (Suez Canal, Panama Canal, and the Magellan, Gibraltar, Torres and Bab el-Mandeb Straits) have long rerouting distances (ranging from 5000 km to around 12,000 km), with detours taking a week or more (Fig. 2c). The highest EVTD is found for the Bab el-Mandeb Strait (USD58.3 billion per year, 0.23% of global trade), Suez Canal (USD44.2 billion per year, 0.18% of global trade) and the Panama Canal (USD11.0 billion per year, 0.04% of global trade). For these chokepoints, the EVTD is driven by some low-probability but high-impact events, such as interstate conflict at the Bab el-Mandeb Strait and Suez Canal (~USD40 billion per year, 0.16% of global trade), drought at the Panama Canal (USD4.3 billion per year, 0.02% of global trade), and armed conflict near the Bab el-Mandeb Strait (USD15.2 billion per year, 0.06% of global trade). However, also high probability events, such as piracy attacks at Bab el-Mandeb Strait, can contribute to the EVTD (USD2.8 billion per year, 0.01% of global trade). The relative EVTD shows that countries in the Middle East, Central Asia and Northern and Western Africa face the high relative risk (Fig. 2d). This is mainly driven by the concentration of flows through some of the most at-risk chokepoints, particularly the Suez Canal (for Central Asia countries) and the Bab al-Mandeb Strait (for countries in the Northern Africa, Horn of Africa and Yemen). For around 76 countries analysed, this relative EVTD is higher than 1% of maritime trade, while for some countries the relative EVTD is 3% or more (Guinea-Bissau, Guinea, Mali, Eritrea, Russia, Sierra Leone and Chad). As such, many countries are facing the risk of long rerouting, which can cause rerouting costs, delays and possible inventory shortages.

The remaining 14 chokepoints have relatively short rerouting distances (< 5000 km) (Fig. 2e). The highest EVTD is found for the Taiwan Strait (USD37.3 billion per year, 0.15% of global trade), Malacca Strait (USD12.8 billion per year, 0.05% of global trade) and Korea Strait (USD8.2 billion per year, 0.03% of global trade). Here the EVTD is driven by high probability but lower impact events, such as tropical cyclone impacting the Taiwan Strait (USD23.5 billion per year, 0.09% of global trade) and Korea Strait (USD6.8 billion per year, 0.03% of global trade), and piracy risk that historically threatened the Malacca Strait (USD9.7 billion per year, 0.04% of global trade). Still, interstate conflict affecting Taiwan Strait (USD13.2 billion per year, 0.05% of global trade) is another important driver of the chokepoint's EVTD. 15 countries have more than 1% of their maritime trade value exposed to disruptions, including Sierra Leone, Iraq, and South Korea, mainly because of their large trade dependency on the Taiwan and Malacca Strait (Fig. 2f). However, disruptions at these chokepoints may only cause minor economic losses given the relative ease to reroute goods cost- and time-efficiently.

The EVTDs at the remaining chokepoints are low on a global scale. Some chokepoints, such as the Cape of Good Hope and Magellan Strait, do not face any hazard risk considered. The Dover and Gibraltar Strait, despite facilitating around 15% of global maritime trade value, have an EVTD of less than one billion per year, given limited hazard

occurrence. For the remaining chokepoints, the smaller size of trade flowing through them in combination with relatively limited hazard exposure results in low EVTD on a global scale. However, disruptions at these chokepoints can still be important for specific countries disproportionately relying on them (Supplementary Fig. 6 for dominant chokepoint per country). For instance, the Luzon Strait is the dominant driver of the EVTD for the island of Tonga, while the Windward Passage is the dominant driver of the EVTD for several Caribbean islands (Haiti, Dominican Republic, Jamaica, Bahamas and the Turks and Caicos Islands).

## Chokepoint economic risk

The economic consequences of chokepoint disruptions can be considerable for long-lasting disruptions. Our analysis of the economic risk of chokepoint disruptions includes the economic impacts associated with delays (due to the development of vessel queues or detours), additional rerouting costs (because of additional operational costs), insurance premiums (to cover transiting chokepoints that face high terrorist, conflict, and piracy risk), revenue losses (to the Suez and Panama Canal Authority), and trade losses in case delays exceed inventory levels (see Methods). The occurrence of different loss types is mainly determined by the rerouting availability (see previous section) in combination with the characteristics of the hazard events, particularly the disruption duration. On the one hand, for short-duration disruptions, only minor delays may occur. On the other hand, for long-duration disruptions, the rerouting costs and possible trade losses may quickly amplify the economic loss. Per chokepoint, we quantify these annual expected economic losses (i.e., economic risk) and allocate them to the countries that face these losses (see Methods). We also present the second-order economic impacts of chokepoint disruptions as a result of an increase in shipping freight rates. Such price spikes occur when rerouting decisions take place, which is effectively taking out vessel capacity from the global supply, impacting freight rates (see Methods and Supplementary Fig. 10 for calibration). In the analysis, we do not consider modal shifts to air or land, as switching trade to alternative transport modes would be capacity-constrained, at least in the short-run. However, for chokepoints with no (maritime) rerouting alternatives and chokepoints with long rerouting alternatives, some modal substitution will likely occur. The feasibility of modal substitution is shaped by the origin and destination of trade (whether alternatives are availability), the available capacity on alternative modes of transport, and commodity characteristics (for some commodities, substitution may be more or less feasible). Hence, our estimates should be considered an upper bound.

Globally, the economic risk is estimated to amount to USD10.7 billion per year (0.04% of global trade), equivalent to the Gross Domestic Product of a country like Mauritania, Malawi or Namibia. The economic risk vary by several orders of magnitude across chokepoints. Nine chokepoints have a high economic risk (>100 million USD per year, see Fig. 3a), with the largest share of risk attributed to the Bab el-Mandeb Strait (USD4.2 billion per year, 0.02% of global trade), the Suez Canal (USD 2.0 billion per year, 0.01% of global trade) and the Malacca Strait (USD2.0 billion per year, 0.01% of global trade). While for the Bab el-Mandeb Strait and Suez Canal, the different economic loss factors all contribute to the total economic risk, for the Malacca Strait, high insurance premiums due to piracy events are driving the economic risk. These three chokepoints alone contribute to 77% of the global economic risk. While the Strait of Hormuz and Bosporus Strait have relatively low EVTD, the lack of alternatives still make it a high-risk chokepoint in terms of economic risk (USD0.4 billion per year and USD0.2 billion per year, respectively, <0.01% of global trade). On the other hand, while the EVTD for the Taiwan Strait is large, the economic risk is considerably smaller (USD0.9 billion per year, <0.01% of global trade) given shorter detours in case of disruptions. Six chokepoints are considered having a medium economic risk (USD10-100 million USD

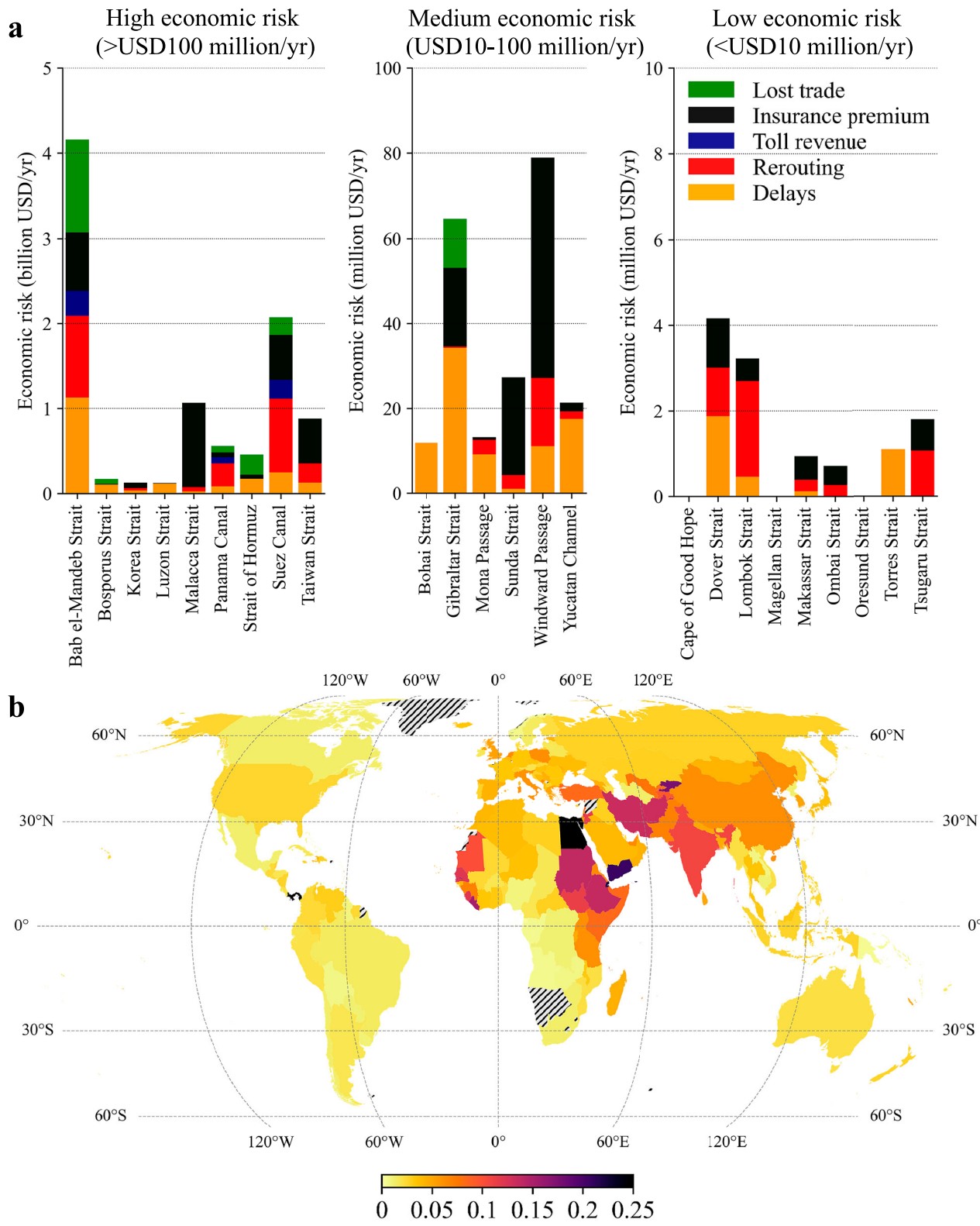

**Fig. 3 | Economic risk due to maritime chokepoint disruptions. a** The economic risk for each maritime chokepoint, broken down by the economic loss type. The chokepoints are grouped in three categories for readability; high ( > USD100 million per year); medium (USD10-100 million per year) and low ( < USD10 million per year). **b** The economic risk as a percentage of the total maritime trade value of a country, highlighting countries with the largest relative economic risk. The hatched pattern indicates countries not included in the analysis. The basemap is from GADM.

per year, Fig. 3a), while the remaining chokepoints have a low economic risk (< USD10 million per year, Fig. 3a).

At a country-level, in relative terms (Fig. 3b), the economic risk is high for those countries relying most on chokepoints with high economic risk (Supplementary Fig. 7 shows the economic risk relative to the total trade value). The countries with the highest relative economic risk are Panama and Egypt (0.3% and 0.6%, respectively), given the large toll revenue losses in case of disruptions at their respective canals. 20 countries have a relative economic risk of over 0.1%, concentrated in Western and Eastern Africa, the Middle East and South Asia, mainly because of their dependency on the Taiwan Strait, Suez Canal, and Bab el-Mandeb Strait (Supplementary Fig. 6).

We estimate a further USD3.4 billion per year (0.01% of global trade) of economic impacts because of increasing container freight rates due to rerouting decisions. These price spikes will harm all economies equally (in relative terms), given that they extend beyond the routes directly impacted by a disruption. The economic impact associated with the freight rate increase is mainly driven by disruptions at the Bab al-Mandeb Strait (42.8%) and Suez Canal (38.0%) (see Supplementary Table 1 for all results). This is because of the large share of global fleet capacity that goes via these two chokepoints and long rerouting distances, resulting in a relatively large share of the global fleet capacity being removed from the market, which is causing such freight rate spikes.

### Joint chokepoint disruptions

A hazard can trigger another hazard, such as an armed conflict increasing the likelihood of a terrorist or piracy attack, so-called compound hazards. In addition, a single hazard can affect multiple chokepoints simultaneously (e.g., a large cyclone event) or can spill over to a larger area (e.g., a local armed conflict in one region causing conflict somewhere else), so-called co-occurring hazards. Such compounding and co-occurring hazards, also called complex hazards[15], have not been considered when estimating the trade and economic risk. Here, we test whether such complex hazards occur in our dataset (see Methods). For the compound hazard analysis, we only consider the four hazards for which we expect some potential triggering occurrence (armed conflict, piracy, terrorist attack and interstate conflict), while for the co-occurrent hazard analysis, we consider five hazards for which we expect potential spatial correlations (cyclones, armed conflict, piracy, terrorist attack and interstate conflict).

In terms of compound hazards, we find three instances, out of the 144 combinations assessed, where two hazards at specific chokepoints show significant statistical dependency (see Supplementary Table 2). In other words, if one occurs, it is likely that the other also occurs. For the Bab el-Mandeb Strait, Bosporus Strait and Lombok Strait, we find a dependency between armed conflict and terrorist attacks, likely because the one was in response to the other.

In terms of co-occurring hazards between the four nonclimatic hazards considered, we find seven instances, out of the 4416 combinations assessed, where two hazard types exhibit a spatial dependency between at least two chokepoints (see Supplementary Table 3). In six out of the seven cases, this relates to piracy incidents showing spatial dependencies across chokepoints, indicating that the rise of piracy attacks in one place may spark attempts elsewhere. Moreover, we find that cyclones can affect multiple chokepoints at once. Out of the 59,273 cyclones events that trigger a closure (given the 10,000 synthetic years evaluated, see Methods), 40% of the events affect more than one chokepoint at the same time. More specifically, 37.5% of events affect exactly two chokepoints, 2.7% affect three and 0.2% affect four maritime chokepoints (see Supplementary Table 4 for details). In other words, simultaneous chokepoint disruptions due to cyclones are common, particularly for chokepoints in close proximity to each other and located in the cyclone belt.

## Discussion

There has been a longstanding recognition of the importance of maritime chokepoints, particularly in terms of their vital role for energy and food security. However, recent events, including the 2023–2024 drought affecting the Panama Canal and 2023–2025 Houthi rebel attacks in the Red Sea, have raised concerns about the potential systemic risks that originate from maritime chokepoint disruptions. The risk is indeed considerable, as a total of USD192 billion of trade each year (0.77% of global trade) is exposed to disruption at the 24 chokepoints that we have analysed. However, we have demonstrated how the significance of economic losses associated with maritime chokepoints disruptions depend upon a combination of factors: (i) the overall value of trade being shipped through the chokepoint (ii) the dependence of any particular country's trade on shipping through the chokepoint (iii) the likelihood of disruptive hazards materialising at the chokepoint and (iv) the costs associated with delays, price spikes, rerouting and inventory shortages.

Whilst the Taiwan Strait, the Strait of Gibraltar and the Malacca Strait are the most important maritime chokepoints in terms of maritime trade value flowing through them, on a country-level we observe a clear geographical clustering, with countries being more dependent on maritime chokepoints near their country's ports. This is particularly true for countries bordering enclosed seas like the Black Sea, the Baltic Sea and the Persian Gulf. Nonetheless, in most cases, countries depend on multiple chokepoints for their trade. For instance, Russia, Kazakhstan and Tajikistan all depend on six chokepoints that each facilitate over 25% of the countries' maritime trade value, making them susceptible to disruptions at any one of them. Countries may not be completely aware of their exposure to disruptions at chokepoints, particularly those located in other continents. Our analysis provides insights into these critical transport dependencies.

Using both empirical and model-based data, we have estimated the probability of eight hazards (cyclones, drought, earthquake, terrorist attacks, armed conflict, interstate conflict, blockages, piracy) occurring at each chokepoint. When trade flow data is combined with the probability of chokepoint disruptions, the most critical chokepoint and hazard combinations are identified as: geopolitical tensions at the Taiwan Strait, Strait of Hormuz and Bab el-Mandeb Strait, droughts at the Panama Canal, and piracy at the Malacca Strait. These dominant hazards resemble recent geopolitical developments in the Taiwan Strait[16] and the Bab el-Mandeb Strait[10], as well as recent droughts affecting the Panama Canal, especially during El-Niño years[17]. We find that countries in the Middle East, Central Asia, Western Africa and the Horn of Africa have 2–3% of their maritime trade value facing potential disruptions each year. These risks arise from low probability, but high impact disruptions, which are often not included in conventional risk management strategies, potentially underestimating risk altogether.

Chokepoint disruptions often happen in isolation. However, in some cases, we find evidence that the occurrences of hazards are related to each other. For instance, we find that armed conflict and terrorism tend to co-occur at specific chokepoints (Lombok Strait, Bosporus Straits, Bab el-Mandeb Strait), and that 40% of tropical cyclones affect more than one chokepoint. Similarly, we also find evidence that piracy attacks in one chokepoint increases the likelihood of attacks elsewhere, even remote chokepoints. These statistical dependency structures warrant further investigation, including understanding the underlying dynamics that explain these dependencies. This is particularly important for those chokepoints where joint disruptions can severely limit rerouting options. Therefore, scenario analysis of multiple chokepoint disruptions should be considered in future work to stress-test the maritime transport system to the most extreme situations, but informed by their joint probability of occurrence.

The economic risk associated with chokepoint disruptions is around USD10.7 billion per year (0.04% of global trade), driven by economic losses at the Suez Canal, Bab-el Mandeb Strait, and the Malacca Strait. The highest relative economic risk is concentrated in a handful of countries (e.g., Panama, Egypt, Yemen, Iraq), which will have most to lose from chokepoint disruptions. On top of that, the global economy could face another USD7.6 billion loss per year (0.03% of global trade) due to container freight rate spikes that make shipping costs more expansive and affect all countries irrespective of their dependencies on the chokepoints that cause these spikes (particularly the Suez Canal and Bab el-Mandeb Strait). The economic losses modelled can cause high-order losses to economies. For instance, the trade losses we model can cause wider supply-chains losses[18] as inventory shortages, leading to production interruptions, can propagate to dependent firms downstream in the supply-chain. Moreover, elevated freight rates can drive up inflation. One study has estimated that a one standard deviation increase in shipping costs can increase headline inflation by around 0.2 percentage points, and double that in island economies[19]. As such, our economic loss estimates should be interpreted as lower bound estimate. On the other hand, we assume that freight blockages are not rerouted over land or via air, and trade losses are not substituted from elsewhere, which can mitigate some of the modelled losses. While we have attempted to improve the characterisation of economic losses from chokepoint disruptions, further research is required for each of the six loss components considered. This should include empirical studies to gain new insights into the magnitude of these losses during past disruptions, as well as simulation-based studies to refine the parametrisation of these losses in model-based risk assessments like ours.

Our modelling provides essential insights into the systemic risks faced by the maritime transport system, which can ripple through and affect countries around the world. Our research underlines how geopolitical unrest could cause large economic burden to countries through transport and trade dependencies, especially for countries in the Middle East and Africa. On top of that, freight rate spikes due to chokepoint disruptions can hurt economies globally, irrespective of their reliance on hazard-prone chokepoints. The latter means that reducing the likelihood of chokepoint disruptions, for instance through strategic security efforts or international cooperation, is in the global interest.

In a period of heightened geopolitical and climate-related risks, our modelling framework may help inspire strategies to build resilience within global supply-chains networks. Specifically, our research highlights that business resilience strategies should be tailored to the specific risks that firms face, given their dependency on a set of chokepoints. For instance, disruptions to chokepoints that require small detours can be effectively buffered using inventories and emergency strategies. However, for chokepoints where disruptions lead to major detours or delays, substitution of production sites or transport modes can be considered. In the most extreme cases, which have inherent low probability of occurrence, specialist insurance products are particularly appropriate. As such, a layered resilience approach is needed, guided by quantitative risk assessments like the one performed in this study.

## Methods

### Maritime chokepoint selection

In this paper, we define maritime chokepoints as strategic maritime canals, passages, channels or straits within maritime routes where large flows of maritime trade converge. We select 24 chokepoints based on the classification of maritime chokepoints in earlier work[20]. However, we make some noticeable differences; (i) we do not use the Dardanelles Strait as separate chokepoint given the proximity to the Bosporus Strait, (ii) we do not select the Bering Strait as chokepoint given limited maritime trade flows through this chokepoint, and (iii)

we add the Bohai Strait and Ombai Strait given their strategic importance for China and Indonesia, respectively. This leaves us with 24 chokepoints, which are shown in Supplementary Fig. 1.

### Maritime trade flows through chokepoints

To derive the dependencies between the maritime chokepoints and the flows of imports and exports of specific countries, we use output of the OxMarTrans global maritime transport model[1]. The OxMarTrans first utilises a global modal split to estimate the share of maritime trade for every bilateral trade flow in the BACI trade database (on a HS6 level) and disaggregates these maritime trade flows to administrative regions (Administrative Region 1, resulting in around 3500 subnational regions globally). A global maritime transport model, informed by observed vessel movements between 1400 ports globally (to set port-to-port connectivity and capacities) is then used for the flow allocation of each bilateral trade flow, capturing the port of export, possible transhipment ports, and port of import, including the maritime routes taken. We then track, for each bilateral trade flow, whether a maritime chokepoint is passed. This can be multiple chokepoints per trade flow (e.g., trade from China to the Netherlands will pass the Taiwan Strait, Malacca Strait, Bab el-Mandeb, Suez Canal, Gibraltar Strait and Dover Strait). Doing so across all maritime trade flows allows us to capture the quantity and value of maritime trade that flows through the various chokepoints, as well as linking this to the countries these trade flows belong to. The OxMarTrans output in ref. 1 was for the 2015 base year. Here, we update the output to use 2022 as base year by scaling the bilateral trade flows (per sector) from 2015 trade quantities and value to 2022 quantities and values. This is based on data in the BACI trade database[21]. It should be noted that our global transport model, utilising a least-cost flow allocation algorithm, may not be able to capture all factors that shape routing decisions in complex hinterlands, in particular for landlocked countries with access to multiple port ranges (e.g., landlocked countries in South America, Europe and Central Asia). Hence, for such countries, the chokepoint dependencies modelled should be interpreted with some care, as modelling errors could be present.

To validate the model output for this exercise, we compare our modelled chokepoint throughput estimates for 2022 with the total vessel capacity that flows through the chokepoints based on Automatic Identification System (AIS), extracted from the PortWatch platform (https://portwatch.imf.org/). The correlation coefficient between the modelled and observed data is 0.78, indicating a good alignment and providing confidence in our modelled throughput estimates.

### Hazard data

We consider eight hazard types, include five human-induced hazards (piracy, armed conflict, terrorist attacks, blockages, and interstate conflict) and three natural hazards (earthquakes, drought and cyclones). For each hazard, we create a spatial event dataset, which captures the geolocation of the historical events and relevant characteristics of each event (e.g., duration, severity). This allows us, per hazard ($h$) and chokepoint ($c$), to derive the following three indicators that we use for the risk quantification: probability ($P$), severity ($S$) and duration ($D$). The hazard probability ($P$) for each event is defined as the annual exceedance probability:

$$P_{c,h} = \frac{1}{T_h}$$

With $T$ being the number of years in the hazard spatial event dataset. The hazard severity is a value between 0 and 1 indicating the severity of the flow duration. A value of 1 indicates that the chokepoint is fully blocked, whereas a value of 0 indicates no flow disruption. The hazard duration is the number of days a chokepoint is in a state of

disruption, ranging from 0 to 365 days. Below, we describe per hazard how these three indicators were derived.

We use 46 years of nautical piracy event data (1978–2024) from the Anti-shipping Activity Messages (ASAM) database, compiled by the National Geospatial-Intelligence Agency (NGA)[22]. This data includes the locations and accounts of hostile acts against ships and mariners. To evaluate the probability, we estimate the annual frequency of events within the chokepoint area, assuming a 50 km buffer around the events, resulting in 820 piracy attacks. The piracy events are shown in green in Supplementary Fig. 9. In line with research evaluating the impacts of piracy events on shipping[14], we assume that every piracy event results in 0.5% less trade for a period of 30 days.

Armed conflict events refer to any incident where armed force was used by an organised actor against another organized actor, or against civilians[23]. Although these conflict events may not happen at sea, conflict events close to the chokepoints can spillover to sea, creating maritime security threats or disrupt port operations in and around the chokepoint. We use the UNCP Georeferenced Event Dataset of conflict events[23]. We select only specific events of interest; (i) conflict events that are classified as intrastate conflict, and (ii) events that have caused at least 10 fatalities. This to exclude interstate conflict, which we treat separately, and use only major conflict events in the frequency analysis, as they are more likely to affect maritime security. The dataset covers events between 1989 and 2023 (35 years), and we use a 50-kilometre buffer to evaluate probability of conflict events within the chokepoint vicinity, leaving us with 96 armed conflict events (see the red markers in Supplementary Fig. 9). We hereby assume that a major armed conflict event will affect a chokepoint for ten days, disrupting 20% of its trade.

Similar as with the armed conflict events, terrorist attacks in the vicinity of chokepoints indicate the propensity for maritime terrorism risk. It has long been hypothesized that terrorist attacks to maritime transport are a becoming more likely, in particular when targeting the maritime routes important for global energy supply[24]. To characterise the probability, we extract previous terrorist events from the Global Terrorism Database[25], which covers events from 1970 to 2020. We retain only relevant events, including those that have killed >5 people (defined here as major events) and those that covered bombing/explosion, hijacking and facility/infrastructure attacks (see blue markers in Supplementary Fig. 9). Using again a 50-kilometre buffer around the events leaves us with 39 relevant past attacks. We assume that a terrorist attack will affect a chokepoint for ten days, disrupting 20% of trade flowing through it.

The risk of blockage because of maritime incidents is related to the width of the chokepoint. We assume that chokepoints with a width of more than 800 m (approximately twice the length of the largest container vessel) do not face the risk of a disruption, as a maritime accident is not likely to cause a disruption of flows. This therefore only includes the Bosporus Strait, the Suez Canal and the Panama Canal. We evaluate the risks of blockages as the number of major shipping incidents happening in and around these three chokepoints. Major shipping incidents here include those classified as 'marine incidents' and 'very serious marine casualty' in the Marine Casualties and Incidents database of the Global Integrated Shipping Information System[26], an information platform of the International Maritime Organization. The database includes accidents between 2006 and 2024 (see orange markers in Supplementary Fig. 9), of which there are 12 in the three chokepoints considered. We assume that a major incident blocks a chokepoint entirely for three days. This is in line with the disruption of the Suez Canal after the Evergreen ship got stuck in it in 2021.

The interstate conflict indicator captures interstate disputes that result in escalating impacts to the maritime chokepoint. We use data from the Militarized Interstate Dispute (MID) database (v5.0), which has data on MID events between 1816 and 2014[27]. A MID is defined as a sequence of related militarized incidents, each a response to one or multiple previous ones. Disputes are always between two or more interstate system members and are non-routine incidents that are governmentally authorized actions. We only include events happening over the last 50 years in the data records (from 1974 to 2014) and those with hostility level defined as 'Display of force', 'Use of force' and 'War' to exclude events that only included threats to use force. The black markers in Supplementary Fig. 9 show the resulting subset of events. Each event has a representative event duration, which we use as the disruption duration. We assume that a disruption reduces trade flow through the chokepoint by 10% for event classified as 'Display of force', 50% for events classified as 'Use of force' and 100% for 'War' events. We take a 200 km buffer around the geographical location recorded in the database, given the large spatial impact region of interstate conflict events and intersect this with the location of the chokepoints. This results in 114 interstate conflicts found in the vicinity of the chokepoints.

Drought is only a relevant risk for the Panama Canal, which is critically dependent on water availability for the operations of its locks. The Panama Canal has faced multiple droughts in the last decade that impeded operations and transits, including in 2014, 2016 and more recently in 2023–2024. We take the latest drought event as reference event. This drought event affected the Canal for 250 days, reducing trade volume through the canal by a third. Recent analysis has shown that the probability of an event like this occurring is around once in 40 years[17], which we take as the representative probability (1/40 per year).

Earthquake risk is also only a relevant risk for the Panama Canal, as it is in the vicinity of two known faults: the Pedro Miguel Fault and Limon Fault[28]. While an earthquake event has not yet happened, the Canal Authority has designed the dams and abutments for a 2500-year earthquake, with a corresponding peak ground acceleration of 0.97g[28]. We take this design criteria as our event probability. In line with the risk classification of the Panama Canal Authority[29], we assume that a devastating earthquake will affect the Canal for a year, blocking trade flows altogether.

Tropical cyclones can cause extreme weather at sea, including high wind, waves and rain, that disrupt the safe navigation of vessels. Here, we use the synthetic cyclone event set of Bloemendaal et al.[30], capturing 10,000 of synthetic cyclone event globally. We track all cyclone events of category-I that were within 250 km of the chokepoint, temporarily disrupting the navigation of vessels through the chokepoint. Per event, we estimate the duration of the disruption based on the number of time periods (per three-hour time slice) that a chokepoint is within the 250 km distance. The median duration is 2.6 days, while the 99th percentile is five days. The event count and annual hit ratio per chokepoint is included in Supplementary Table 5.

## Risk quantification

We use the historical hazard event data to quantify the risk of maritime chokepoint disruptions and the economic loss associated with that. We propose two metrics to evaluate the systemic risk, the expected value of trade disrupted (*EVTD*) and the economic risk (*ER*) per country (*i*).

The *EVTD* (in USD/yr) can be written as follows:

$$EVTD_i = \sum_h \sum_c P_{c,h} \cdot \frac{T_{i,c}}{365} \cdot S_{c,h} \cdot D_{c,h} \tag{1}$$

With *P* the annual probability of occurrence of the specific hazard (*h*), *D* the duration of the disruption (days), *S* the severity of flow disruption (between 0 and 1), and *T* the value of maritime trade of country (*i*) that flows through a specific chokepoint (*c*). This metric captures the systemic risk exposure, i.e., the amount of trade value that has the potential to be affected each year.

The *ER* (USD/yr) is a function of the EVTD but captures the various channels of economic losses. We consider five loss channels:

$$EL_i = EL_{d,i} + EL_{r,i} + EL_{p,i} + EL_{t,i} + EL_{l,i} \tag{2}$$

With $EL_d$ the loss associated with delays, $EL_r$ the loss associated with rerouting, $EL_p$ the loss associated with increased insurance premiums, $EL_l$ the loss associated with trade losses in case delay times exceed inventory levels, and $EL_t$ the loss associated with reduced toll revenues for Panama and Egypt because of reduced flows through their respective canals.

### Rerouting decision

The occurrence of these different loss channels depends critically on the decision to reroute or not during disruptions of a certain duration, if at all feasible (for the Bohai Strait, Strait of Hormuz, Oresund Strait and Bosporus Strait, no rerouting is possible). The decision to reroute depends on the $D_{c,h}$ and the rerouting distance $\Delta d_c$. Rerouting happens if:

$$D_{c,h} > \frac{\Delta d_c}{V} \tag{3}$$

With $V$ being the average vessel speed, set to 16 knots. Rerouting does not happen when:

$$D_{c,h} \leq \frac{\Delta d_c}{V} \ or \ no \ rerouting \ alternative \tag{4}$$

If goods are not rerouted, they are delayed for the duration $D$.

### Economic loss quantification

The economic loss of delays for a country (*i*) can be written as:

$$EL_{d,i} = \sum_h \sum_c P_{c,h} \cdot \frac{I_{i,c}}{365} \cdot S_{c,h} \cdot \sum_j^D VOT \cdot j \quad if \ no \ rerouting \tag{5}$$

$$EL_{d,i} = \sum_h \sum_c P_{c,h} \cdot \frac{I_{i,c}}{365} \cdot S_{c,h} \cdot \sum_j^{\Delta d_c/V} VOT \cdot j \quad if \ rerouting \tag{6}$$

With $I$ the country import value flowing through the chokepoint and *VOT* the value of time of a day of delay (in % per day). The *VOT* is assumed to be 1% of trade value per day of delay, in line with ref. 31. The loss associated with delay scales linearly with the duration (in a 7-day delay, 1 day worth of goods is 7 days delayed, 1 day worth of goods is 6 days delayed, etc.).

The economic loss associated with rerouting is only borne in case rerouting is taking place. The costs accrue to the importing country as the additional costs of longer routes (e.g., additional fuel and crew expenditure) is added to the transport costs:

$$EL_{r,i} = 0 \quad if \ no \ rerouting \tag{7}$$

$$EL_{r,i} = \sum_h \sum_c P_{c,h} \cdot \frac{IQ_{i,c}}{365} \cdot S_{c,h} \cdot \Delta d_c \cdot Cd \cdot D_{c,h} \quad if \ rerouting \tag{8}$$

With *IQ* the import quantity (in volume terms) and *Cd* the maritime costs per t·km shipped, which is set to USD0.003 per t·km[1]. The import quantity is an output of the OxMarTrans model.

During the occurrence of specific hazard that can impact cargo or vessels directly, the insurance premiums that shippers pay on their cargo and vessel value can rapidly spike. For instance, the Russia-Ukraine war spiked insurance premiums for ships crossing the Black

Sea to 1–3%[32,33], while ships deciding to cross the Bab el-Mandeb Strait during the Houthi rebel attacks faced insurance premiums of 0.6–2%[34,35]. Historically, vessels crossing regions with high piracy rates faced elevated insurance premiums in the order of 0.1–0.3%[36]. We characterise the loss due to these insurance rates as:

$$EL_{p,i} = \sum_h \sum_c P_{c,h} \cdot \frac{I_{i,c}}{365} \cdot (1 - S_{c,h}) \cdot D_{c,h} \cdot IP_h \tag{9}$$

With *IP* the increase in insurance premium. We set the *IP* value to 1% for the interstate conflict and 0.1% for piracy, terrorist and armed conflict. As can be seen, the loss is only attributed to the fraction of trade that still crosses the chokepoint (*1-S*). We use the value of imports ($I_{i,c}$) of any given country flow through a specific chokepoint to quantify the insurance loss to that country, as the importing economy pays for insurance fees and, hence, faces the loss due to an increase of the insurance rate.

We further add the revenue loss to the Panama and Suez Canal Authority to the estimates:

$$EL_{t,i} = \sum_h \sum_c P_{c,h} \cdot \frac{R_{c,i}}{365} \cdot S_{c,h} \cdot D_{c,h} \tag{10}$$

with *R* the annual toll revenue and *i* being Egypt for the Bab El-Mandeb and Suez Canal, and Panama for the Panama Canal. The annual revenue is set to USD5 billion per year for the Panama Canal and USD8 billion per year for the Suez Canal.

Both trade delays and rerouting decisions can cause inventories to run out, especially for firms that have adopted just-in-time inventory management practises. Here, we assume that if importing firms run out of inventories, they must stop production and face losses worth the value of imports. This therefore, does not take higher-order supply-chain losses into account. Inventories levels differ considerably per sector, country and even firms within sectors. Here, we split the total trade into three groups (*g*) with an associated inventory level ($T_{inv}$). We do this by approximating the fraction of firms (*f*) with a given inventory based on the distribution of firm-level inventory data for French firms in Lafrogne-Joussier et al.[37] (see Supplementary Table 6). The inventory levels set also align with average sector-specific inventory levels reported for the United Kingdom in Pichler et al.[38]. We can write the trade losses as:

$$EL_{l,i} = \sum_g \sum_h \sum_c P_{c,h} \cdot \frac{f_g \cdot I_{i,c}}{365} \cdot S_{c,h} \cdot \left( D_{c,h} - T_{inv,g} \right)$$
$$if \ no \ rerouting \ and \ D > T_{inv} \tag{11}$$

$$EL_{d,i} = 0 \quad if \ no \ rerouting \ and \ D < T_{inv} \tag{12}$$

$$EL_{l,i} = \sum_g \sum_h \sum_c P_{c,h} \cdot \frac{f_g \cdot I_{i,c}}{365} \cdot S_{c,h} \cdot \left( \frac{\Delta d_c}{V} - T_{inv,g} \right)$$
$$if \ rerouting \ and \ \frac{\Delta d_c}{V} > T_{inv} \tag{13}$$

$$EL_{d,i} = 0 \ if \ rerouting \ and \ \frac{\Delta d_c}{V} < T_{inv} \tag{14}$$

In other words, losses occur when the delay or rerouting duration exceeds the inventory level, with the economic loss being the value of production that cannot be generated during this period.

### Shipping price increase

Apart from the direct economic losses due to maritime chokepoint disruptions, as discussed above, there may also be an indirect economic loss due to rising freight rates (i.e., the price paid to ship goods).

This rise in freight rates is attributed to the fact that the rerouting of vessels takes out a certain capacity from the total available fleet capacity, i.e., a supply shock. Freight rates have been shown to spike due to such supply shocks, for instance because of the rerouting of ships due to the Houthi rebel attacks (2023-2025). A global shipping price increase will affect every country reliant on shipping, irrespective if they are directly affected by the chokepoint disruption.

Here, we derive a simple approach to estimate the shipping price increase as a function of the fleet capacity 'lost' due to rerouting. This approach follows as the same rationale as the approach by Arvis et al.[39], who relates deviations in the shipping price index to share of the vessel fleet that is affected by a disruption. Given a global fleet, that could (in theory) be reallocated based on changes supply and demand patterns, we can estimate a single coefficient for all chokepoint disruptions, similar as proposed in Arvis et al.[39]. We calibrate the model on the increase in container freight rates during the period 2023-2024, which saw such elevated prices (e.g., dry bulk and tanker prices spiked less). To do so, we first derive the monthly ($m$) reduction in daily container fleet capacity ($C$) going through the Suez Canal and Panama Canal ($c$) for the period 2023–2024 compared the monthly values in 2022.

$$\Delta C_{m,c} = 1 - \frac{C_{m,c,2023-2024}}{C_{m,c,2022}} \tag{15}$$

We do this based on data provided in the PortWatch platform (https://portwatch.imf.org/). We then construct a metric of the monthly supply shock ($S$), equal to the reduction in monthly fleet capacity as a fraction of the total fleet capacity and the respective delay (Panama) and rerouting (Suez) days ($D$):

$$S_m = \sum_c \Delta C_{m,c} \cdot \frac{C_{m,c}}{C_m} \cdot D_c \tag{16}$$

With $D$ set to 15 days for Panama (the average delay observed during the drought restrictions) and 12 days for Suez (the average rerouting duration around the Cape of the Good Hope). We then fit a linear regression between the $S_m$ and the freight cost increase ($\%FC$) using the Drewry World Container Index ($R^2 = 0.74$), as shown in Supplementary Fig. 10. We then apply this approach to the modelled disruptions, replacing $D$ with the rerouting duration, if rerouting is taking place. Using this approach, we predict the $\%FC$ per disruption event, after which we can estimate the total economic risk due to freight rate ($EL_f$):

$$EL_f = \sum_h \sum_c P_{c,h} \cdot \frac{IQ_{container} \cdot d}{365} \cdot D_{c,h} \cdot Cd \cdot \%FC_{c,h} \tag{17}$$

with $IQ$ the global container volume imported and $d$ the average kilometres shipped per ton of goods. $d$ is set to 9500 km per ton based on the work of Verschuur et al.[1], who estimated that globally 9.4 billion tonnes of maritime trade is moved 90.5 trillion tonnes·km in 2015. We proxy $IQ$ based on the global container fleet capacity (329.5 million deadweight tons) versus the global total maritime fleet capacity (2.35 billion deadweight tons) based on UNCTAD (2024)[40]. Hence, we set IQ to 14% of the total maritime import volume.

**Joint disruption probability**

For the above risk calculations, we assume that all hazard-induced chokepoint disruptions are independent from each other. It is possible that hazards have some level of dependence at the chokepoint level (i.e., the occurrence of one hazard can trigger another one) and across chokepoints (i.e., one hazard can affect multiple chokepoints simultaneously). We therefore evaluate whether this assumption, which is necessary to make the risk calculation tractable, is indeed justified. Earthquake and drought hazards are not considered here, given that

they only affect one chokepoint (Panama Canal). Similarly, blockage events are also excluded for the analysis, given that we consider them random in nature.

To test the dependency between hazard-induced disruptions, we perform two sets of further analysis. First, at each chokepoint, we test whether there is a statistically significant relationship between two hazards occurring within the same year. We only do this for the four hazards where we expect such as potential relationship: armed conflict, piracy, terrorist attacks and interstate conflict. For all overlapping years between each combination of hazard, we classify whether a given year faced a hazard or not (binary 1 or 0). We then run Fisher's exact test[41] to test whether binary time series are considered independent or not by calculating the contingency table between hazard incidences. In total, this results in 6 combinations (($4 \times 3$)/2) per chokepoint, so 144 possible combinations in total. Given the sparsity of events, we deem a joint-occurrence of events to be statically significant if (i) we find a $p$-value smaller than 0.05, and (ii) joint-occurrence is observed at least twice. In Supplementary Table 2, we report all the statistically significant combinations of hazards affecting a specific maritime chokepoint.

Second, we test whether there is a statistical dependence between hazards across chokepoints. This could be because an event can affect two chokepoints at the same time, such as a cyclone event, or because of some other spatial interdependency (e.g., an armed conflict spilling over to another region). For cyclone events, we evaluate the number of times an event affects multiple chokepoints, within the synthetic event dataset we use (see 'Tropical Cyclones' section). The results are reported in Supplementary Table 4 (see below). For armed conflict, piracy, terrorist attacks and interstate conflict events, we again analyse the joint occurrence based on the binary time series, but now across chokepoints and across hazards (including similar hazards). Per hazard combination considered, this yields 276 chokepoint combinations ($24 \times 23/2$), totally 4416 (($4 \times 4$) x 276) combinations. Here, we define a joint-occurrence of events to be statistically significant if (i) we find a $p$-value smaller than 0.05, and (ii) the joint incidence occurs at least five times. In Supplementary Table 3, we report all the statistically significant combinations of hazards across different chokepoints.

## Data availability

All input data is publicly available at the sources cited in the paper. The country dependencies on chokepoints and data needed to reproduce the analysis are available in a Zenodo repository[42].

## Code availability

The code to reproduce the analysis is stored in a Zenodo repository[42].

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

## Acknowledgements

This research has been conducted as part of the Oxford Martin School Programme on Systemic Resilience, for which JV, JL and JWH received funding. We are grateful for support provided by GallagherRe for this research through the Gallagher Research Centre.

## Author contributions

J.V. conceptualised the research, performed the analysis and led the writing of the manuscript. J.L. performed the joint probability analysis and contributed to the writing of the manuscript. J.W.H. conceptualised the research, provided input and contributed to the writing of the manuscript.

## Competing interests

The authors declare no competing interests.
