## [Transparent Peer Review file · Nature Communications]

Systemic impacts of disruptions at maritime chokepoints

Corresponding Author: Dr Jasper Verschuur

Version 0:

Reviewer comments:

Reviewer #1

(Remarks to the Author)

The paper deals with a topic of strategic commercial importance as it assesses the potential disruption effects of the main chokepoints of maritime circulation. This is a relevant analysis that would warrant publication and the following are my observations and comments.

Line 15. "Yet, the exposure of countries to these disruptions is not known, inhibiting adequate preparedness." This may be an overstatement. These potential effects are known but not comprehensively assessed and, in many cases, unclear.

Line 24. "Countries most exposed to these economic risks are located in the Middle-East, Africa and Latin America." This may be too generic. Please be more specific and provide some examples.

Line 51. The issue of climate change is becoming hyperbolic, taking almost a religious tone, and applied as an "explanation" for many events. In this regard, the drought in Panama was not severe by historical standards as there were periods when rainfall was even lower. The issue is much more one of water demand than of water supply through rainfall. The canal expansion, the growth in the number of transits and general urban demand are much more significant factors. The paper discusses the effects of cyclones and droughts and not climate change.

Figure 3a. It would be relevant to state why the shares on the figure add up above 100% since it may confuse so readers. Since many of these passages are in a sequence along maritime trade routes, the summations are "autocorrelated".

Figure 3b. It is a very confusing figure that is difficult to read, particularly because of the color classification. The classification can be considered a bit fallacious since many chokepoints, such as the Taiwan Strait, can be bypassed, while others cannot. So, I do not believe that China has a trade dependency on the Taiwan Strait since there is an important transpacific orientation. Same for Northern Europe as it relates to the Dover Strait (still a higher deviation). There could be a way around this perceived fallacy, which would require ordering the chokepoints by deviation into two main classes. A first classification can be done for dependency for primary chokepoints with no deviation (Panama, Suez, Gibraltar, Suez, Mandeb, Oresund, Bosphorus, Hormuz). Then, as a subset, a subclassification can be done with secondary chokepoints with deviation. This may be a good way to depict the hierarchy of dependency which appears to be missing from the analysis.

Line 216. "For instance, the Luzon Strait is the dominant driver of the EVT D for the island of Tonga, while the Windward Passage is the dominant driver of the EVT D for several Caribbean islands (Haiti, Dominican Republic, Jamaica, Bahamas and the Turks and Caicos Islands). I would argue that this does not make too much sense because of the short deviations involved, which would mean no EVT D of significance, at least not related to these chokepoints.

Line 272. A nuance concerning Panama. It has ports on both the Pacific and the Atlantic side, paradoxically implying that its imports and exports would not be impacted by a disruption of its own canal (maybe indirectly through less shipping connectivity). However, the loss of revenue could be substantial. Still, the capacity shortages that took place until mid-2024 did not impact the canal revenue to a large extent as rates were increased, and a bidding process was put forward to see who was willing to pay to use the more limited transit capacity. The same could be said about Egypt, which has ports on both the Med and Red Sea sides; so it would internally be less impacted by a closure of its own canal (Egypt is of course substantially impacted by the Red Sea crisis/Bab el-Mandeb). Could it be argued, paradoxically, that Panama and Egypt have low EVT D because of this characteristic, even though these values can be potentially much higher for third countries? In some ways, the paper could be interpreted as an argument for reshoring.

Line 357. "In a period with heightened risks of geopolitical conflict and climate change, our modelling framework provides the basis for a better understanding the implications of these risks, and help build resilience within the ever-increasing complexity of global supply-chains networks." How? I do not think the paper offers resilience strategies. It discusses disruptions, not mitigation.

Line 382. "There is no generally agreed set of maritime chokepoints, nor a consistent definition." Yes, there is. The maritime chokepoints are very well-defined, and ample literature on the issue is available. The issue is more about their relative strategic importance and if more marginal ones, such as Bohai, should be considered.

Reviewer #2

(Remarks to the Author)

This study provides a comprehensive analysis, and the content is well-organized. The article creatively introduces quantitative models using AIS data to explore the disruption risks of maritime chokepoints. It calculates the probabilities and severity of various risk events, offering valuable insights into the shipping industry and countries' dependences on these chokepoints. It also evaluates the economic impact of disruptions to shipping trade volume and value. Overall, it is an engaging and meaningful contribution to the field.

I have some comments below for the author to consider:

1. It is assumed that disruptions at chokepoints occur independently (lines 183–185). However, multiple disruptions may occur simultaneously and interact with one another. Could the authors provide a more detailed rationale for this independence assumption and discuss how this assumption might influence their results?
2. The study appears to compare human-made disruptions with natural disasters. Given that natural disasters (e.g., typhoons, floods) are external events while armed and interstate conflicts are human-induced and often closely associated with trade dynamics, I wonder if these two kinds of disruptions are comparable. In Figure 2(a), the expected trade disruption values for human-made events are substantially higher than those for natural disasters, with some maritime chokepoints showing nearly zero expected impact. A similar issue is in Figure 3(a), where several chokepoints have nearly zero estimated economic risk. Could the authors clarify how differences in scale and underlying mechanisms between these disruption types are addressed in the analysis.
3. Considering that global shipping operates as a closely interconnected network, it is possible that a disruption at one chokepoint could have spillover effects on others, potentially leading to chain reactions. The current analysis does not appear to explicitly consider these network interdependencies. Could the authors discuss whether and how spillover or cascading effects are incorporated into their model or provide insight into how such network effects might influence the overall findings?
4. Line 436 formula (3) part 2, parameter C_d is not explained in the context.
5. Line 436 formula (3) part 2, the author introduced VPT "to convert the value shipped to quantity", what's the meaning and dimension of the calculating result? Besides, the dimensions of the two parameters ($\Delta d/V_{Dc,h}$) are both in terms of time, and does the resulting dimension after multiplication remain consistent?
6. Line 438 formula (3), part 1, the author explained in Line 446 "the vessel are delayed for the duration...". Part 3, the author explained in Line 455 "trade being lost given no maritime transport alternatives". Since part 1 assumes the vessels would not reroute under the condition of no transport alternatives (part 3), why are the trades lost instead of being delayed?

Version 1:

Reviewer comments:

Reviewer #1

(Remarks to the Author)

The author has addressed the majority of my comments.

Risk and disruption values have limited meaning when presented in absolute numbers since the reader (most likely) does not know the frame of reference. I would suggest that in each case, the number is presented, followed by a frame of reference such as "(x% of the global GDP)". A frame of reference is occasionally presented, but systematically placing these frames of reference helps better convey the message, particularly to the non-academics who will source the paper. Or maybe present the value as a ratio (share of GDP) instead of an absolute value.

In Figure 1b, I fail to understand why Bolivia is dependent on the Magellan Strait. There is minimal trade going through this chokepoint, and Bolivia's trade, as a landlocked country, exits through Peru/Chile or Uruguay/Paraguay/Brazil. Technically, landlocked countries should inherit the chokepoint of the country they are the most reliant on to access maritime trade. Maybe a short nuance to no rerouting to be added. In this case, the trade flows in question would need to shift entirely to other modes (rail, road, air), which is not a possibility in terms of available capacity. Still, a portion (unknown) could shift. Line 280 addresses the issue, but this almost only relates to non-rerouting situations.

(Remarks on code availability)

Reviewer #2

(Remarks to the Author)

The authors have made extensive additions and revisions in the new version of the paper. An analysis of the joint checkpoint disruption phenomenon has been added, along with a discussion on the correlation between compound hazards

and co-occurring hazards. Furthermore, the presentation of FIG 2 and FIG 3 has been improved, offering a clearer depiction of EVT distribution across different regions by distinguishing between various types of checkpoints.

The paper redesigns the risk quantification model, breaking down the economic losses at the national and regional levels into categories such as losses from rerouting, delays, increased insurance premiums, toll revenue reductions, and trade losses caused by inventory shortages. Additionally, a new model has been introduced to calculate the total economic risk resulting from freight rate increases. However, I still have the following comments for authors' consideration.

Comments :

1. There are numerous expression errors in Economic Risk Metric Section. Line 676-678 "ELd,i" should be "ELr,i". Line 718-721 "ELd,i", "Tint".
2. Formula 9, why use li,c (explained in Line 665 "import value flowing through the chokepoint") to calculate losses due to insurance rate.
3. In Formula 16, the monthly supply shock is derived solely from delay days (Panama) and rerouting days (Suez). This metric is then used in a regression model to estimate the freight rate increase (%FC). Why can this model generalize to all the checkpoints and hazards?
4. How can the authors validate the rationality and reliability of the economic risk quantification model? Are there any research methods used as references, or is there authoritative data to verify the model's calculation results? The model includes numerous approximate parameters, such as Line 759: "set to 30% of the total import volume" and "set to 10,000 km per ton." How can the reasonableness of such parameters be substantiated?

(Remarks on code availability)

Version 2:

Reviewer comments:

Reviewer #2

(Remarks to the Author)

I am satisfied with the revisions and appreciate the authors' efforts.

(Remarks on code availability)

REVIEWER COMMENTS

Reviewer #1 (Remarks to the Author):

The paper deals with a topic of strategic commercial importance as it assesses the potential disruption effects of the main chokepoints of maritime circulation. This is a relevant analysis that would warrant publication and the following are my observations and comments.

We would like to thank the reviewer for their engagement with our paper and overall positive feedback. However, the reviewer also outlines some useful points for further improvement of the paper. Below, we have tried to address these comments to the best of our ability.

Line 15. "Yet, the exposure of countries to these disruptions is not known, inhibiting adequate preparedness." This may be an overstatement. These potential effects are known but not comprehensively assessed and, in many cases, unclear.

We agree that this was an overstatement without adding the nuance as suggested by the reviewer. We have adjusted the text, see L17-18:

“Yet, the exposure of countries to these disruptions is not comprehensively assessed, inhibiting adequate preparedness “

Line 24. "Countries most exposed to these economic risks are located in the Middle-East, Africa and Latin America." This may be too generic. Please be more specific and provide some examples.

We have rewritten this sentence in line with the updated results.

Line 51. The issue of climate change is becoming hyperbolic, taking almost a religious tone, and applied as an "explanation" for many events. In this regard, the drought in Panama was not severe by historical standards as there were periods when rainfall was even lower. The issue is much more one of water demand than of water supply through rainfall. The canal expansion, the growth in the number of transits and general urban demand are much more significant factors. The paper discusses the effects of cyclones and droughts and not climate change.

We have purposely not made statement whether past events were caused by climate change or not, exactly for the reasons the reviewer outlined. However, climate change is an important discussion point, as it may increase the frequency (and already has in some cases) of certain extremes, including the Gatun Lake shortages. However, this is obviously not the sole, or even primary, driver. Hence, we try not to push for a narrative in which climate change will be the main cause of chokepoint disruptions, where in fact, humans are most to “blame”.

Figure 3a. It would be relevant to state why the shares on the figure add up above 100% since it may confuse so readers. Since many of these passages are in a sequence along maritime trade routes, the summations are "autocorrelated".

Many thanks for pointing this out. This is indeed due to the sequential transits through multiple chokepoints. We have added the following sentence to clarify this, see L126-128.

“Trade flows also often cross multiple chokepoints, resulting in the sum of the share of trade across chokepoints being larger than 100%.”

Figure 3b. It is a very confusing figure that is difficult to read, particularly because of the color classification. The classification can be considered a bit fallacious since many chokepoints, such as the Taiwan Strait, can be bypassed, while others cannot. So, I do not believe that China has a trade dependency on the Taiwan Strait since there is an important transpacific orientation. Same for Northern Europe as it relates to the Dover Strait (still a higher deviation). There could be a way around this perceived fallacy, which would require ordering the chokepoints by deviation into two main classes. A first classification can be done for dependency for primary chokepoints with no deviation (Panama, Suez, Gibraltar, Suez, Mandeb, Oresund, Bosphorus, Hormuz). Then, as a subset, a subclassification can be done with secondary chokepoints with deviation. This may be a good way to depict the hierarchy of dependency which appears to be missing from the analysis.

We agree with this point. We decided to display the figure like this to add one more step before going into the economic risk calculations which included the rerouting calculations. But we also agree this now creates a potential fallacy. We have tried to change this figure and indeed split the chokepoint into three categories; (i) no rerouting, (ii) long rerouting length (>5000 km) and (iii) low to moderate rerouting length (<5000 km). This also sets the basis for understanding Figure 3. We have therefore also adjusted the result section to reflect those changes, see L197-266, and changed Figure 2..

Line 216. "For instance, the Luzon Strait is the dominant driver of the EVTD for the island of Tonga, while the Windward Passage is the dominant driver of the EVTD for several Caribbean islands (Haiti, Dominican Republic, Jamaica, Bahamas and the Turks and Caicos Islands). I would argue that this does not make too much sense because of the short deviations involved, which would mean no EVTD of significance, at least not related to these chokepoints.

We have rewritten this text in line with the earlier comment made.

Line 272. A nuance concerning Panama. It has ports on both the Pacific and the Atlantic side, paradoxically implying that its imports and exports would not be impacted by a disruption of its own canal (maybe indirectly through less shipping connectivity). However, the loss of revenue could be substantial. Still, the capacity shortages that took place until mid-2024 did not impact the canal revenue to a large extent as rates were increased, and a bidding process was put forward to see who was willing to pay to use the more limited transit capacity. The same could be said about Egypt, which has ports on both the Med and Red Sea sides; so it would internally be less impacted by a closure of its own canal (Egypt is of course substantially impacted by the Red Sea crisis/Bab el-Mandeb). Could it be argued, paradoxically, that Panama and Egypt have low EVDT because of this characteristic, even though these values can be potentially much higher for third countries?

In some ways, the paper could be interpreted as an argument for reshoring.

We agree with the point that a disruption of its own canal, does not necessarily impede shipping directly. The dominance is indeed because of the revenue losses incurred (we do not capture the Panama Canal Authority's bidding procedures, which indeed buffered some of the revenue losses). It could indeed be the case that they have flexibility to ship goods via

alternative ports. However, capacity of ports on either side of the respective canals may not be enough to take over all trade, but indeed this is not captured in our analysis.

Line 357. "In a period with heightened risks of geopolitical conflict and climate change, our modelling framework provides the basis for a better understanding the implications of these risks, and help build resilience within the ever-increasing complexity of global supply-chains networks." How? I do not think the paper offers resilience strategies. It discusses disruptions, not mitigation.

While it discusses disruptions, not mitigation, our quantitative framework can help provide the basis for resilience options. For instance, it can (i) help quantify the economic benefits of choosing alternative routes (if possible) that are longer but avoid high-risk chokepoints, (ii) it can help quantify the economic benefit of improved security of certain chokepoints with high geopolitical or conflict risk, (iii) it can help inform firms that trade across borders identify suitable resilience strategies to deal with these low probability, but high impacts risks (e.g., larger inventories, supply diversification).

We have now added a clearer description how we believe our analysis can help build supply-chain resilience, see L439-448.

Line 382. "There is no generally agreed set of maritime chokepoints, nor a consistent definition." Yes, there is. The maritime chokepoints are very well-defined, and ample literature on the issue is available. The issue is more about their relative strategic importance and if more marginal ones, such as Bohai, should be considered.

This was indeed not well framed, we have removed this sentence.

Reviewer #2 (Remarks to the Author):

This study provides a comprehensive analysis, and the content is well-organized. The article creatively introduces quantitative models using AIS data to explore the disruption risks of maritime chokepoints. It calculates the probabilities and severity of various risk events, offering valuable insights into the shipping industry and countries' dependences on these chokepoints. It also evaluates the economic impact of disruptions to shipping trade volume and value. Overall, it is an engaging and meaningful contribution to the field.

We would like to thank the reviewer for their engagement with our paper and overall positive feedback. However, the reviewer also outlines some useful points for further improvement of the paper. Below, we have tried to address these comments to the best of our ability.

I have some comments below for the author to consider:

1. It is assumed that disruptions at chokepoints occur independently (lines 183–185). However, multiple disruptions may occur simultaneously and interact with one another. Could the authors provide a more detailed rationale for this independence assumption and discuss how this assumption might influence their results?

We indeed had to make the necessary assumption that chokepoint disruptions are independent to make the calculations tractable. Upon reflection, we have added some further analysis on the likelihood of joint disruptions, see L330-359, which illustrate that some dependency is indeed present, but only in some exceptional cases. However, this requires further research as well.

Moreover, we have added further discussion on the cascading impacts of multiple chokepoint disruptions (L401-411).

2.The study appears to compare human-made disruptions with natural disasters. Given that natural disasters (e.g., typhoons, floods) are external events while armed and interstate conflicts are human-induced and often closely associated with trade dynamics, I wonder if these two kinds of disruptions are comparable. In Figure 2(a), the expected trade disruption values for human-made events are substantially higher than those for natural disasters, with some maritime chokepoints showing nearly zero expected impact. A similar issue is in Figure 3(a), where several chokepoints have nearly zero estimated economic risk. Could the authors clarify how differences in scale and underlying mechanisms between these disruption types are addressed in the analysis.

While we agree that there are indeed fundamental differences between the different types of hazards, we believe it is still suitable to bring them all together in one framework. This is also one of the main contributions of our paper as previous studies have often focused on single hazards alone, instead of a comprehensive overview how these compare.

There are indeed large differences in the modelled impacts. However, the risks are not zero, but differ an order magnitude. This is driven by changes in the likelihood and duration of the hazards, and hence the resulting impacts. For instance, cyclones happen relatively frequently, but only disrupt chokepoints for a few days. On the other hand, geopolitical conflicts are lower probability, but can affect chokepoints for several months.

We have changed both Figure 2 and Figure 3, so the relative differences are more clear now. Moreover, we have added more reflection why there are such substantial differences between the chokepoints throughout the text.

3.Considering that global shipping operates as a closely interconnected network, it is possible that a disruption at one chokepoint could have spillover effects on others, potentially leading to chain reactions. The current analysis does not appear to explicitly consider these network interdependencies. Could the authors discuss whether and how spillover or cascading effects are incorporated into their model or provide insight into how such network effects might influence the overall findings?

4.Line 436 formula (3) part 2, parameter C_d is not explained in the context.

We have added the explanation of C_d now.

5.Line 436 formula (3) part 2, the author introduced VPT “to convert the value shipped to quantity”, what’s the meaning and dimension of the calculating result? Besides, the dimensions of the two parameters ($\Delta d/V D_{c,h}$) are both in terms of time, and does the resulting dimension after multiplication remain consistent?

After reflecting on this formula, we realised the dimensions indeed do not add up. We have come up with an alternative formulation that also deals with point 6 raised. We have made substantial changes to all the loss functions, so that they are consistent and more in line with observed (anecdotal) impacts during past chokepoint disruptions.

6.Line 438 formula (3), part 1, the author explained in Line 446 “the vessel are delayed for

the duration...”. Part 3, the author explained in Line 455 “trade being lost given no maritime transport alternatives”. Since part 1 assumes the vessels would not reroute under the condition of no transport alternatives (part 3), why are the trades lost instead of being delayed?

After reflecting on this formula, we have come to the conclusion that there are indeed some inconsistencies in the approach. We have tried to resolve this using the new approach followed, see L637-761. We have broken down the impact analysis per loss contributor, and also added insurance premiums as additional loss components, as well as an first-order estimate of the shipping price spike.

Reviewer #1 (Remarks to the Author):

The author has addressed the majority of my comments.

We would like to thank the reviewer (again) for their useful and constructive comments. While the major comments have been addressed, a few (minor) comments remained. We have now tried our best to capture these comments in the revised text, as well as through our detailed reply as indicated below.

Risk and disruption values have limited meaning when presented in absolute numbers since the reader (most likely) does not know the frame of reference. I would suggest that in each case, the number is presented, followed by a frame of reference such as "(x% of the global GDP)". A frame of reference is occasionally presented, but systematically placing these frames of reference helps better convey the message, particularly to the non-academics who will source the paper. Or maybe present the value as a ratio (share of GDP) instead of an absolute value.

Thank you for this suggestion. We have added the relative impacts figures throughout now. Given we are talking about trade, we use global trade value as reference, which was 25 trillion in 2022 (our reference trade year).

In Figure 1b, I fail to understand why Bolivia is dependent on the Magellan Strait. There is minimal trade going through this chokepoint, and Bolivia's trade, as a landlocked country, exits through Peru/Chile or Uruguay/Paraguay/Brazil. Technically, landlocked countries should inherit the chokepoint of the country they are the most reliant on to access maritime trade.

It is indeed correct that Bolivia trades most of their goods through Chilean and Peruvian ports (less so through Uruguay/Paraguay/Brazil). Whether Bolivia has the same chokepoint dependencies as Peru and Chile, however, depends on the destination of trade of the respective countries.

We looked into the transport model output to search for an explanation. See here the reasons:
-Chile exports more with North America and Canada, and receives imports from Spain and Germany, all through the Panama Canal. Chile also depends on the Magellan Strait for trade with partners like Brazil and Argentina, some of which is trade via land but also a large share is trade via maritime.

-Peru exports large values to the US+Canada and Europe (Great Britain, Netherlands, Spain, Switzerland), which explain the large dependency on Panama. Moreover, they are more dependent (more than Chile and Bolivia) on the Taiwan Strait given large trade with China.

-Bolivia on the other hand, depends almost equally on the Panama Canal (10.1% of trade value), driven by trade with countries like the United States and Columbia, and Magellan Strait (10.7%), given larger trade with Argentina and Brazil.

The transport model we rely on is a least-cost transport allocation model, whilst in practice some logistical decisions may be based on other factors, in particular in complex hinterland transport regions such as Latin America. Moreover, it does not include inland water transport (besides the Panama canal) as an alternative mode of transport, which may explain the larger than expected maritime trade flow from Bolivia to Argentina (some of which would go over the Paraguay-Parana river system. In 2019, around 1.3 million tonnes of Bolivian goods

where exported by river, out of 5.6 million tonnes total exports, according to the follow article:

<https://en.unav.edu/web/global-affairs/bolivia-reduce-su-dependencia-de-los-puertos-de-chile-con-alternativas-en-peru-y-la-via-fluvial-atlantica>

Hence, some of these exports may have gone via inland water instead of maritime. Still, some of those goods, in particular low value bulky goods may have been loaded at Argentinian ports for exports to Asia via the Magellan. This highlights, once again, the complexity of certain routing decisions and their dependencies on certain trade flows.

Still we admit that our estimates may have some model uncertainties/error, which we want to further highlight. Hence, we have added a sentence to reflect this in LX-X:

“It should be noted that our global transport model, utilising a least-cost flow allocation algorithm, may not be able to capture all factors that shape routing decisions in complex hinterlands, in particular for landlocked countries with access to multiple port ranges (e.g., landlocked countries in South America, Europe and Central Asia). Hence, for such countries, the chokepoint dependencies modelled should be interpreted with some care, as modelling errors could be present.”

Maybe a short nuance to no rerouting to be added. In this case, the trade flows in question would need to shift entirely to other modes (rail, road, air), which is not a possibility in terms of available capacity. Still, a portion (unknown) could shift. Line 280 addresses the issue, but this almost only relates to non-rerouting situations.

This is well noted. It could indeed be the case that a (small fraction) would be shifted for the no rerouting case, but equally for the longer rerouting situations. For instance, for a completely closure of the Suez Canal, some trade may be rerouted on the Asia to Europe railway lines, although we would be speaking about smaller percentages given the order magnitude difference in trade flowing through these two alternatives.

See our edits in L290-295:

“However, for both chokepoints with no (maritime) rerouting alternatives and chokepoints with long rerouting alternatives, some modal substitution will likely occur. The feasibility of modal substitution is shaped by the origin and destination of trade (whether alternatives are availability), the available capacity on alternative modes, and commodity characteristics (for some commodities, substitution may be more or less feasible). Hence, our estimates should be considered an upper bound.”

Reviewer #2 (Remarks to the Author):

The authors have made extensive additions and revisions in the new version of the paper. An analysis of the joint checkpoint disruption phenomenon has been added, along with a discussion on the correlation between compound hazards and co-occurring hazards. Furthermore, the presentation of FIG 2 and FIG 3 has been improved, offering a clearer depiction of EVTD distribution across different regions by distinguishing between various types of checkpoints.

The paper redesigns the risk quantification model, breaking down the economic losses at the

national and regional levels into categories such as losses from rerouting, delays, increased insurance premiums, toll revenue reductions, and trade losses caused by inventory shortages. Additionally, a new model has been introduced to calculate the total economic risk resulting from freight rate increases. However, I still have the following comments for authors' consideration.

We would like to thank the reviewer (again) for their useful and constructive comments. While the major comments have been addressed, a few (minor) comments remained. We have now tried our best to capture these comments in the revised text, as well as through our detailed reply as indicated below.

Comments :

1. There are numerous expression errors in Economic Risk Metric Section. Line 676-678 "ELd,i" should be "ELr,i". Line 718-721 "ELd,i", "Tint".

Thank you for pointing out these typos. This was indeed an oversight. We checked all other equations on accuracy as well. We found one other typo (the index in the toll revenue loss formula, $R_c = R_{c,i}$), which we fixed now as well.

2. Formula 9, why use $I_{i,c}$ (explained in Line 665 "import value flowing through the chokepoint") to calculate losses due to insurance rate.

Thanks for pointing this out. As the importing country pays for shipping insurance, they would be facing the economic loss of increased insurance rates. Hence, why we use the import value of the importing country. We have added a sentence to clarify this:

L725-728

"We use the value of imports ($I_{i,c}$) of any given country flow through a specific chokepoint to quantify the insurance loss to that country, as the importing economy pays for insurance fees and, hence, faces the loss due to an increase of the insurance rate."

3. In Formula 16, the monthly supply shock is derived solely from delay days (Panama) and rerouting days (Suez). This metric is then used in a regression model to estimate the freight rate increase (%FC). Why can this model generalize to all the checkpoints and hazards?

According to other research, the freight rate is a function of supply shock of vessel capacity (see paper by the World Bank, Arvis et al. (2024)). We therefore aimed to construct a similar supply-shock driven model using the situation of the simultaneous disruption of the Panama and Suez Canal as a way to calibrate this model. The coefficient we estimate is, in our view, generalisable to supply shocks to other chokepoints, given that we have a fixed global fleet that could (in theory) be reallocated to any location. In practise, there may be some smaller regional effects, but we assume for simplicity that this is not the case.

We have added this justification to the text, see L759-777.

Reference:

Arvis, J.-F., Rastogi, C., Rodrigue, J.-P. & Ulybina, D. *A Metric of Global Maritime Supply Chain Disruptions: The Global Supply Chain Stress Index (GSCSI)*. (2024).

4. How can the authors validate the rationality and reliability of the economic risk

quantification model? Are there any research methods used as references, or is there authoritative data to verify the model's calculation results?

Unfortunately, there are no authoritative data to verify the risk estimates, given the intrinsic nature of risk as a metric that cannot be measured. Our individual loss calculations are informed by some empirical estimates referenced in the text, although not directly comparable. Other aspects of the model, for example vessel rerouting as a result of the Bab el-Mandeb and Panama disruptions, have been empirically verified with AIS data. The innovative nature of our paper is that we bring this all together within a probabilistic assessment of risk, as well as making several methodological innovations in terms of approximating the different loss elements. We hope that the risk estimates we bring into the world will spark new research aiming at refining them.

We add a line on this in L443-448:

“While we have advanced previous research in terms of characterising the different economic loss components of chokepoint disruptions, further research is needed in each one of the six loss components considered. This includes both empirical studies to gain new insights on the magnitude of these losses during past disruptions, as well as simulation-based studies to refine the parametrisation of these losses to include in model-based risk assessments like ours.”

The model includes numerous approximate parameters, such as Line 759: “set to 30% of the total import volume” and “set to 10,000 km per ton.” How can the reasonableness of such parameters be substantiated?

These two parameters require indeed some further explanation. The 10,000 km per ton was derived based on the results of our previous work, where we estimated that a total of 9.4 billion tonnes of maritime trade moved 90.5 trillion ton-km. To be even more specific, we set the value to 9,500 km per ton. This may vary slightly from country to country, but for simplicity we use a global value.

The IQ figure, we now proxied based on the latest data from UNCTAD on their global vessel capacity number, showing that the container fleet is around 14% (in terms of capacity) of the total maritime vessel fleet. We use this percentage to estimate IQ, setting it to 14% of the total maritime import volume.

See L795-800 that reflect these updates:

“with IQ the global container volume imported and d the average kilometres shipped per ton of goods. d is set to 9,500 km per ton based on the work of Verschuur et al. (2022)¹, who estimated that globally 9.4 billion tonnes of maritime trade is moved 90.5 trillion tonnes-km in 2015. We proxy IQ based on the global container fleet capacity (329.5 million deadweight tons) versus the global total maritime fleet capacity (2.35 billion deadweight tons) based on UNCTAD (2024)⁴¹. Hence, we set IQ to 14% of the total maritime import volume.”